# Risk-taking behaviors in adolescent men who have sex with men (MSM): An association between homophobic victimization and alcohol consumption

**Evette Cordoba**[1]*, **Robert Garofalo**[2,3], **Lisa M. Kuhns**[2,3], **Cynthia Pearson**[4], **Josh Bruce**[5], **D. Scott Batey**[6], **Asa Radix**[7], **Uri Belkind**[7], **Marco A. Hidalgo**[8,9], **Sabina Hirshfield**[10], **Rafael Garibay Rodriguez**[1], **Rebecca Schnall**[1]

1 School of Nursing, Columbia University, New York, New York, United States of America, 2 Division of Adolescent Medicine, Ann & Robert H. Lurie Children's Hospital of Chicago, Chicago, Illinois, United States of America, 3 Department of Pediatrics, Northwestern University, Feinberg School of Medicine, Evanston, Illinois, United States of America, 4 Indigenous Wellness Research Institute, School of Social Work, University of Washington, Seattle, Washington, United States of America, 5 Birmingham AIDS Outreach, Birmingham, Alabama, United States of America, 6 Department of Social Work, University of Alabama at Birmingham, Birmingham, Alabama, United States of America, 7 Callen-Lorde Community Health Center, New York, New York, United States of America, 8 Children's Hospital Los Angeles, The Saban Research Institute, Los Angeles, CA, United States of America, 9 Keck School of Medicine, University of Southern California, Los Angeles, CA, United States of America, 10 Department of Medicine, STAR Program, SUNY Downstate Health Sciences University, Brooklyn, New York, United States of America

* ec2678@cumc.columbia.edu

**Data Availability Statement:** All relevant data are within the paper and its Supporting Information files.

## Abstract

### Objective

The aim of this study was to determine whether homophobic victimization was associated with alcohol consumption and riding with an intoxicated driver or driving a car while under the influence of alcohol or drugs among adolescent men who have sex with men (MSM).

### Methods

Cross-sectional analysis used baseline data from a national HIV prevention trial (NCT03167606) for adolescent MSM aged 13–18 years (N = 747). Multivariable logistic regression models assessed associations between homophobic victimization (independent variable) and alcohol-related outcomes (dependent variables), controlling for age, parents' education level, sexual orientation, health literacy, race, and ethnicity.

### Results

Most participants (87%) reported at least one form of homophobic victimization in their lifetime, with verbal insults being the most frequently reported (82%). In the bivariate analysis, alcohol consumption and riding with an intoxicated driver or driving a car while under the influence were associated with many forms of victimization. Exposure to at least one form of victimization was associated with increased odds of alcohol consumption (OR: 2.31; 95%

**Funding:** Research reported in this publication was supported by the National Institute of Minority and Health Disparities of the National Institutes of Health under award number U01MD011279 awarded to RS and RG, and the National Institute of Nursing Research of the National Institutes of Health under award number K24NR018621 awarded to RS. The content is solely the responsibility of the authors and does not necessarily represent the official views of the National Institutes of Health. The funders had no role in study design, data collection and analysis, decision to publish, or preparation of the manuscript.

**Competing interests:** The authors have declared that no competing interests exist.

CI: 1.38–3.87) and riding with an intoxicated driver or driving a car while under the influence (OR: 2.25; 95% CI: 1.26–4.00), after controlling for covariates.

## Conclusion

Increased risk of alcohol consumption and risky alcohol-related behaviors were found among adolescent MSM who experienced homophobic victimization. Interventions should address homophobic victimization and its impact on adolescent MSM, as well as disentangling motivations for underage drinking, riding with an intoxicated driver or driving a car while under the influence.

## Introduction

Underage alcohol consumption remains a significant public health concern in the United States (US) [1], particularly among sexual and gender minority (SGM) youth [2]. Despite an overall decline in alcohol consumption among underage youth [3], disparities among SGM (i.e., lesbian, gay, bisexual, asexual, transgender) youth still exist [4–6] and have even widened in recent years [2, 6]. SGM youth have a disproportionately higher prevalence of alcohol consumption, binge drinking [5, 7–9], and early initiation of alcohol use compared to their heterosexual peers [9, 10]. Disparities in underage alcohol consumption warrant concern given its association with adverse health behaviors such as risky sexual activities, injuries, suicide attempts, and riding with an intoxicated driver [11]. Moreover, among SGM youth, alcohol-related disparities may persist into late adolescence or adulthood, increasing the risk of heavy drinking and alcohol-related disorders later in life [10, 12]. SGM youth also have steeper drinking trajectories into young adulthood compared to their heterosexual counterparts [13]. To address this health concern, it is imperative for researchers to improve their understanding of risk factors predicting underage drinking among SGM youth in effort to inform targeted prevention strategies and empirically based interventions [14].

Accumulating evidence on alcohol-related disparities among SGM youth suggest that one contributing factor is homophobic victimization [5, 7, 15–17], which is defined as a direct or indirect act of aggression, negative behavior, derogatory comment, verbal harassment, physical threat, or act of violence towards a person due to their actual or perceived sexual orientation [18]. Today, homophobic victimization among youth remains pervasive and occur most often among SGM youth [8, 19–21]. Approximately 69% and 26% of SGM youth report being verbally and physically harassed in school because of their sexual orientation, respectively [22]. Moreover, SGM males report more severe or direct forms of victimization (e.g., physical assault or assault with weapon) compared to SGM females [18, 20]. In response to homophobic victimization, researchers suggest that alcohol consumption among SGM youth is a coping mechanism to help alleviate negative effects or stress induced by victimization [23]. In fact, some studies have found school-based victimization to be strongly related to substance use (including alcohol use) among SGM youth [3, 15]. Future research is needed to identify specific forms of homophobic victimization experienced by SGM youth and their association with alcohol consumption to better understand the risk factors contributing to underage drinking in this population.

Even with well documented studies linking homophobic victimization and alcohol use among SGM youth, there are three major limitations that have not been addressed. First, few studies have examined the effects of different forms of homophobic victimization (i.e., verbal,

physical, or sexual assaults) on underage alcohol consumption in this population [15]. By identifying the forms of homophobic victimization associated with alcohol consumption, interventions can be implemented to target specific victimization tactics aimed at SGM youth. Such information is also essential to fully capture the experiences of SGM youth and how homophobic victimization contributes to the risk of underage drinking. Second, limited research has been conducted to understand victimization as a risk factor for alcohol use specifically among adolescent men who have sex with men (MSM) [3]. We contend that adolescent MSM may be at particularly high risk of underage drinking due to their frequent exposure to severe forms of victimization [18, 20], warranting concern. Additionally, such focus on adolescent MSM alone will allow researchers to compare alcohol consumption among adolescent MSM who experienced homophobic victimization to adolescent MSM who did not, unlike previous comparisons that have been made to heterosexual youth [2, 6, 8]. Furthermore, research on underage drinking among adolescent MSM is needed to improve our understanding of alcohol-related disparities found within the SGM population. Third, to our knowledge, no previous study has assessed the association between homophobic victimization and risky alcohol-related behaviors, specifically driving while intoxicated or riding with an intoxicated driver. Previously, over 20% of SGM youth reported riding with an intoxicated driver and 8% drove while intoxicated [9]. Identifying homophobic victimization as a risk factor for alcohol-related behaviors will provide evidence that victimization may influence adverse health behaviors beyond substance abuse and significantly impacts the health of adolescent MSM. Therefore, the objectives of the present study were two-fold: 1. To measure the prevalence of various forms of homophobic victimization experienced by adolescent MSM using a national sample and 2. To determine whether significant associations exist between various forms of homophobic victimization and underage drinking and risky alcohol-related behaviors among adolescent MSM. We hypothesized that exposure to homophobic victimization increased the risk of underage drinking and risky alcohol-related behaviors in this population.

## Methods

### Study population

A cross-sectional analysis was conducted using baseline data from a randomized control trial (RCT), 'Male Youth Pursuing Empowerment, Education, and Prevention around Sexuality' (MyPEEPS). MyPEEPS is a nationwide mobile-health HIV prevention RCT that was conducted during 2018–2020 using a diverse sample of adolescent MSM (N = 761). Eligible participants were adolescent MSM aged 13–18 years, assigned male at birth, identified as male or non-binary, sexually attracted to males, HIV-negative or unknown status, had access to a smartphone or tablet, felt comfortable speaking and reading English, and resided in the US or US territories. The current analysis used data from 747 study participants, of whom homophobic victimization data were available, and excluded 1.8% (N = 14) of participants with missing data on victimization. All participants provided written informed assent or consent. The study was approved by the Institutional Review Board (IRB) of Columbia University, which served as the single IRB for the MyPEEPS RCT (NCT 03167606).

### Exposure

Homophobic victimization was measured using standard questions on lifetime exposure to 10 forms of victimization based on sexual orientation or attraction (i.e. because they were gay, bisexual, or a guy who has sex with guys): *verbal insult*, *threatened with physical violence*, *threatened with a weapon*, *object thrown at you*, *physical assault*, *attacked sexually*, *threatened to be outed*, *chased or followed*, *property damaged*, and *spat upon* (Cronbach Coefficient α =

0.83). The frequency of each victimization occurrence was reported by participants as 'never', 'once', 'twice', or 'three times or more'. Each response was dichotomized to create a binary independent variable for lifetime homophobic victimization (categorized as 'no, never occurred' or 'yes, occurred at least once').

## Outcomes

Two dichotomous outcome variables were of interest: underage drinking and risky alcohol-related behavior. Participants self-reported underage drinking by answering whether they drank alcohol (more than a few sips but not including alcohol taken during family or religious events) in the past 12 months. Risky alcohol-related behavior was determined by asking participants: *Have you ever ridden in a car driven by someone (including yourself) who was high or had been using alcohol or drugs*? Of the 747 participants included in this study, all had complete data on alcohol consumption. However, we excluded one participant with missing data on whether they have ridden in a car with an intoxicated driver or have driven a car while they were under the influence of alcohol or drugs.

## Covariates

Based on extensive review of the literature and an analysis of a directed acyclic graph (DAG), the following potential covariates were selected a priori: age, health literacy, mother's education level, father's education level, sexual orientation, race, and ethnicity. Participant's age was treated as a continuous variable in the analysis. Health literacy was determined by asking participants their level of confidence when filling out medical forms by themselves. This was categorized as 'not at all/ little bit', 'somewhat', or 'quite a bit/extremely'. All other variables were categorized as follows: mother's and father's education level (advanced graduate degree, graduated 4-year college, some college/ technical training, high school graduate/ GED/ did not finish high school), sexual orientation (bisexual, mostly gay/only gay/homosexual, mostly heterosexual/only heterosexual, or something else), race (American Indian/Alaskan Native, Asian American, Black/African American, Native Hawaiian/Asian Pacific Islander, White/Caucasian, Multiracial, or unknown/not reported), and Hispanic/Latino/Latinx ethnicity (yes or no).

## Data analysis

Univariate analyses were conducted to estimate the prevalence of 10 forms of homophobic victimization. The frequency and percent of participants' characteristics, underage drinking, and riding/driving a car while under the influence were calculated by homophobic victimization. To assess percent difference in the bivariate analysis, the Pearson's chi-squared test or Fisher's exact test was applied as appropriate [24]. Logistic regression models assessed the associations between 1. homophobic victimization and underage drinking and 2. homophobic victimization and riding/driving a car while under the influence. Multivariable logistic regression models adjusted for participant's age, parents' education level, sexual orientation, health literacy, race, and ethnicity. Participants with missing data on covariates were dropped from the multivariable regression models. Odds ratios (ORs), 95% confidence intervals (CIs), and p-values were calculated for all regression models. A p-value of $<0.05$ was considered statistically significant. All analyses were performed using SAS 9.4 software (SAS Institute, Cary, NC, USA).

## Results

The majority of study participants were 16 years or older (72%), had parents who received some college education or more (67% of mothers and 59% of fathers), self-identified as mostly

or only gay/homosexual (75%), had quite a bit of confidence or were extremely confident when filling out medical forms by themselves (54%), self-identified as belonging to a racial minority group (50%), and did not identify as Hispanic/Latino/Latinx (59%). (Table 1)

Most participants (87%) were exposed to at least one form of homophobic victimization in their lifetime, with verbal insults (yelled at or criticized) being the most frequently reported form of victimization (82%). Approximately half of participants reported that someone threatened to out them (tell someone that they are gay or bisexual) (56%) or threatened them with physical violence (46%). One-third of participants had an object thrown at them (33%) and a quarter of them were physically assaulted (punched, kicked, or beaten) (25%). About one in five participants were chased or followed (22%), attacked sexually (21%), or had their property damaged (18%). Sixteen percent of participants were spat upon and 13% were threatened with a knife, gun, or another weapon. (Table 2)

In the bivariate analysis, underage drinking was significantly associated with seven forms of homophobic victimization: verbal insults, threatened with physical violence, threatened with a weapon, attacked sexually, threatened to be outed, property damaged, and being spat upon. Being physically assaulted (punched, kicked, or beaten) was borderline associated with underage drinking. Riding with an intoxicated driver or driving a car while under the influence of alcohol or drugs was significantly associated with all forms of homophobic victimization except being threatened with a knife, gun, or another weapon. (Table 3) Unadjusted regression models, performed to account for the logistic distribution of the data, also confirmed these results.

Lifetime exposure to at least one form of homophobic victimization was significantly associated with greater odds of underage drinking, even after adjusting for age, parents' education level, sexual orientation, health literacy, race, and ethnicity (OR: 2.31; 95% CI: 1.38–3.87). In the multivariable regression models shown in Table 4, five of the 10 forms of homophobic victimizations were associated with underage drinking: verbal insults (OR: 1.99; 95% CI: 1.27–3.10), threatened with physical violence (OR: 1.59; 95% CI: 1.12–2.25), attacked sexually (OR: 1.65; 95% CI: 1.07–2.55), property damaged (OR: 1.63; 95% CI: 1.02–2.60), and being spat upon (OR: 1.75; 95% CI: 1.08–2.85). Similarly, exposure to at least one form of homophobic victimization was significantly associated with riding with an intoxicated driver or driving a car while under the influence of alcohol or drugs, after adjusting for covariates (OR: 2.25; 95% CI: 1.26–4.00). More specifically, in the multivariable models seven of the 10 forms of homophobic victimization were individually associated with greater odds of riding with an intoxicated driver or driving a car while under the influence: verbal insults, physical assault, attacked sexually, threatened to be outed, chased, property damaged, and being spat upon. For each of the seven forms of victimization, if exposed, participants had a significantly greater odds (ORs range: 1.22–2.08) of riding with an intoxicated driver or driving a car while under the influence compared to those who were not exposed to homophobic victimization. Being threatened with a knife, gun, or another weapon was not assessed in a multivariable model for this outcome since there was no significant association found in the bivariate analysis. (Table 5)

## Discussion

The current study contributes new knowledge to the field by providing cross-sectional evidence of various forms of homophobic victimization, alcohol use, and risky alcohol-related behaviors among adolescent MSM.

This study found that exposure to homophobic victimization was common among adolescent MSM, with 87% reporting exposure to at least one form of victimization due to their sexual orientation or attraction. It has been postulated that exposure to homophobic

**Table 1. Characteristics of study participants by exposure to homophobic victimization.**

| Characteristics | Total Sample | Victimization | | p-value |
|---|---|---|---|---|
| | | No | Yes | |
| | N = 747 (%) | N = 94 (%) | N = 653 (%) | |
| **Age at baseline** | | | | 0.6682* |
| 13 | 22 (3.0) | 3 (3.2) | 19 (2.9) | |
| 14 | 82 (11.0) | 8 (8.5) | 74 (11.3) | |
| 15 | 107 (14.3) | 12 (12.8) | 95 (14.6) | |
| 16 | 184 (24.6) | 19 (20.2) | 165 (25.3) | |
| 17 | 206 (27.6) | 30 (31.9) | 176 (27.0) | |
| 18 | 146 (19.5) | 22 (23.4) | 124 (19.0) | |
| **Mother's education level** | | | | 0.0702 |
| Advanced graduate degree | 109 (15.8) | 13 (14.4) | 96 (16.0) | |
| Graduated 4-year college | 178 (25.8) | 21 (23.3) | 157 (26.2) | |
| Some college/ technical training | 174 (25.3) | 16 (17.8) | 158 (26.4) | |
| High school graduate/ GED | 134 (19.5) | 20 (22.2) | 114 (19.0) | |
| Did not finish high school | 94 (13.6) | 20 (22.2) | 74 (12.35) | |
| Missing | 58 | | | |
| **Father's education level** | | | | 0.4953 |
| Advanced graduate degree | 92 (15.1) | 12 (15.2) | 80 (15.1) | |
| Graduated 4-year college | 128 (21.1) | 12 (15.2) | 116 (21.9) | |
| Some college/ technical training | 139 (22.9) | 16 (20.3) | 123 (23.3) | |
| High school graduate/ GED | 149 (24.5) | 24 (30.4) | 125 (23.6) | |
| Did not finish high school | 100 (16.5) | 15 (19.0) | 85 (16.1) | |
| Missing | 139 | | | |
| **Sexual orientation** | | | | 0.0007* |
| Bisexual | 158 (21.2) | 26 (27.7) | 132 (20.2) | |
| Mostly or only gay/ homosexual | 563 (75.4) | 63 (67.0) | 500 (76.6) | |
| Mostly or only heterosexual | 6 (0.8) | 2 (2.1) | 4 (0.6) | |
| Something else | 20 (2.7) | 3 (3.2) | 17 (2.6) | |
| **Health literacy** | | | | 0.7324 |
| Not at all or a little bit | 156 (20.9) | 22 (23.4) | 134 (20.5) | |
| Somewhat | 183 (24.5) | 24 (25.5) | 159 (24.4) | |
| Quite a bit or extreme | 408 (54.6) | 48 (51.1) | 360 (55.1) | |
| **Race** | | | | 0.0217* |
| American Indian/ Alaskan Native | 42 (5.6) | 4 (4.3) | 38 (5.8) | |
| Asian American | 72 (9.6) | 17 (18.1) | 55 (8.4) | |
| Black/African American | 151 (20.2) | 22 (23.4) | 129 (19.8) | |
| Native Hawaiian/ Asian Pacific Islander | 10 (1.3) | 1 (1.1) | 9 (1.4) | |
| White/Caucasian | 280 (37.5) | 27 (28.7) | 253 (38.7) | |
| Multiracial | 95 (12.7) | 7 (7.5) | 88 (13.5) | |
| Unknown/not reported | 97 (13.0) | 16 (17.0) | 81 (12.4) | |
| **Hispanic/Latino/Latinx ethnicity** | | | | 0.7144 |
| No | 440 (58.9) | 57 (60.6) | 383 (58.7) | |
| Yes | 307 (41.1) | 37 (39.4) | 270 (41.4) | |

*Fishers exact test was used (need to check p-values) when cell number<5.

**Table 2. Exposure to homophobic victimization in lifetime.**

| Homophobic Victimization | N (%) |
|---|---|
| **Total** | **747 (100)** |
| Exposure to at least one form of homophobic victimization | 653 (87.4) |
| No exposure to homophobic victimization | 94 (12.6) |
| **10 Forms of Homophobic Victimization*** | |
| Verbally insulted (yelled at, criticized) | 614 (82.2) |
| Someone threatened to out you | 418 (56.0) |
| Threatened with physical violence | 341 (45.7) |
| Had an object thrown at you | 243 (32.5) |
| Been punched, kicked, or beaten | 186 (24.9) |
| Someone chased or followed you | 166 (22.2) |
| Attacked sexually | 153 (20.5) |
| Your property was damaged | 137 (18.3) |
| Been spat upon | 119 (15.9) |
| Threatened with a knife, gun, or another weapon | 100 (13.4) |

*Categories are not mutually exclusive.

victimization among adolescent MSM may act as a sexual minority-specific stressor, increasing the risk of adverse health behaviors notably at earlier ages compared to heterosexual counterparts [10, 12, 15, 25]. According to the minority stress model, experiencing stressors related to sexual minority status can increase the likelihood of alcohol misuse and early alcohol initiation among SGM [14, 26–28]. Our study provides evidence that alcohol consumption is common among adolescent MSM, with half of the adolescent MSM reporting alcohol consumption, which warrants concern given that they are all under the US legal drinking age of 21. We found that among adolescent MSM underage drinking was associated with various forms of homophobic victimization, such as verbal insults or threats, physical or sexual assaults, and property damage. We compared our results to studies focused on school-based victimization and alcohol use among SGM youth due to the lack of data surrounding homophobic victimization in MSM youth in other setting. Data from one study demonstrated that alcohol-related disparities in SGM youth could be explained by exposure to school-based victimization [17]. Another large study of over 13,000 Midwestern high school students found that homophobic teasing was associated with alcohol use among all students regardless of sexual identity, however, the association was strongest among SGM [29]. Conversely, another study found that victimization did not fully explain alcohol-related disparities when comparing White, Black, and Latino SGM youth [30], while another study reported no evidence to support the association between victimization and alcohol consumption [31]. Our study expanded on previous work by examining specific forms of homophobic victimization. We found five forms of victimization associated with alcohol consumption, which were verbal insults, being spat upon, sexual attacks, property damaged, and being threatened with physical violence. Although these forms of victimization had similar effects on alcohol consumption (ORs 1.6–2.0), there was some gradient found. For instance, verbal insults had a stronger association with alcohol consumption than other forms of victimization such as being threatened with physical violence or having property damaged. This finding may inform targeted interventions to focus on specific forms of homophobic victimization to significantly reduce underage alcohol consumption among adolescent MSM. To our knowledge, this is the first study to examine specific forms of

**Table 3. Exposure to specific forms of homophobic victimization by underage drinking and riding with an intoxicated driver or driving while under the influence of alcohol or drugs.**

| Victimization | | Underage drinking | | | Riding/ driving under the influence | | |
|---|---|---|---|---|---|---|---|
| | Total Sample N (%) | No | Yes | | No | Yes | |
| | | N (%) | N (%) | p-value | N (%) | N (%) | p-value |
| Total | 747 | 356 (47.7) | 391 (52.3) | | 496 (66.4) | 250 (33.5) | |
| **Verbally insulted (yelled at, criticized)** | | | | 0.0028 | | | 0.0060 |
| No | 133 (17.8) | 79 (22.2) | 54 (13.8) | | 102 (20.6) | 31 (12.4) | |
| Yes | 614 (82.2) | 277 (77.8) | 337 (86.2) | | 394 (79.4) | 219 (87.6) | |
| **Threatened with physical violence** | | | | 0.0041 | | | 0.0420 |
| No | 406 (54.4) | 213 (59.8) | 193 (49.4) | | 283 (57.1) | 123 (49.2) | |
| Yes | 341 (45.7) | 143 (40.2) | 198 (50.6) | | 213 (42.9) | 127 (50.8) | |
| **Had an object thrown at you** | | | | 0.1251 | | | 0.0213 |
| No | 504 (67.5) | 250 (70.2) | 254 (65.0) | | 349 (70.4) | 155 (62.0) | |
| Yes | 243 (32.5) | 106 (29.8) | 137 (35.0) | | 147 (29.6) | 95 (38.0) | |
| **Been punched, kicked, or beaten** | | | | 0.0486 | | | 0.0085 |
| No | 561 (75.1) | 279 (78.4) | 282 (72.1) | | 387 (78.0) | 173 (69.2) | |
| Yes | 186 (24.9) | 77 (21.6) | 109 (27.9) | | 109 (22.0) | 77 (30.8) | |
| **Threatened with a knife, gun, or another weapon** | | | | 0.0121 | | | 0.5711 |
| No | 647 (86.6) | 320 (89.9) | 327 (83.6) | | 432 (87.1) | 214 (85.6) | |
| Yes | 100 (13.4) | 36 (10.1) | 64 (16.4) | | 64 (12.9) | 36 (14.4) | |
| **Attacked sexually** | | | | 0.0003 | | | 0.0026 |
| No | 593 (79.5) | 303 (85.1) | 290 (74.4) | | 409 (82.6) | 183 (73.2) | |
| Yes | 153 (20.5) | 53 (14.9) | 100 (25.6) | | 86 (17.4) | 67 (26.8) | |
| issing | 1 | | | | | | |
| **Someone threatened to out you** | | | | 0.0360 | | | 0.0002 |
| No | 329 (44.0) | 171 (48.0) | 158 (40.4) | | 243 (49.0) | 86 (34.4) | |
| Yes | 418 (56.0) | 185 (52.0) | 233 (59.6) | | 253 (51.0) | 164 (65.6) | |
| **Someone chased or followed you** | | | | 0.0502 | | | 0.0060 |
| No | 581 (77.8) | 288 (80.9) | 293 (74.9) | | 401 (80.9) | 180 (72.0) | |
| Yes | 166 (22.2) | 68 (19.1) | 98 (25.1) | | 95 (19.2) | 70 (28.0) | |
| **Your property was damaged** | | | | 0.0200 | | | 0.0126 |
| No | 610 (81.7) | 303 (85.1) | 307 (78.5) | | 418 (84.3) | 192 (76.8) | |
| Yes | 137 (18.3) | 53 (14.9) | 84 (21.5) | | 78 (15.7) | 58 (23.2) | |
| **Been spat upon** | | | | 0.0320 | | | 0.0185 |
| No | 628 (84.1) | 310 (87.1) | 318 (81.3) | | 428 (86.3) | 199 (79.6) | |
| Yes | 119 (15.9) | 46 (12.9) | 73 (18.7) | | 68 (13.7) | 51 (20.4) | |
| **Any victimization** | | | | 0.0017 | | | 0.0035 |
| No | 94 (12.6) | 59 (16.6) | 35 (9.0) | | 75 (15.1) | 19 (7.6) | |
| Yes | 653 (87.4) | 297 (83.4) | 356 (91.0) | | 421 (84.9) | 231 (92.4) | |

homophobic victimization and alcohol consumption in this population, and future investigation is needed to understand the causal mechanisms linking victimization to underage drinking.

In addition, our study found that 34% of participants have ridden with an intoxicated driver or driven a car while under the influence of alcohol or drugs. This is of great public health concern given that young drivers are more likely to be involved in alcohol-related crashes when driving under the influence [32]. Our estimates, however, were higher than that previously reported by the Centers for Disease Control and Prevention (CDC) Morbidity and Mortality

**Table 4. Bivariate and multivariable models assessing the association between exposure to homophobic victimization and underage drinking.**

| | Underage drinking | | | | | |
| --- | --- | --- | --- | --- | --- | --- |
| | Bivariate | Multivariable* | Multivariable* | Multivariable* | Multivariable* | Multivariable* |
| | OR (95% CI) | OR (95% CI) | OR (95% CI) | OR (95% CI) | OR (95% CI) | OR (95% CI) |
| **Verbally insulted** | 1.78 | 1.99 | | | | |
| | (1.22, 2.60) | (1.27, 3.10) | | | | |
| **Threatened with physical violence** | 1.53 | | 1.59 | | | |
| | (1.14, 2.04) | | (1.12, 2.25) | | | |
| **Been punched, kicked, or beaten** | 1.40 | | | 1.41 | | |
| | (1.00, 1.96) | | | (0.93, 2.14) | | |
| **Threatened with weapon** | 1.74 | | | | 1.60 | |
| | (1.12, 2.69) | | | | (0.94, 2.73) | |
| **Attacked sexually** | 1.97 | | | | | 1.65 |
| | (1.36, 2.85) | | | | | (1.07, 2.55) |
| **Someone threatened to out you** | 1.36 | 1.28 | | | | |
| | (1.02, 1.82) | (0.91, 1.80) | | | | |
| **Your property was damaged** | 1.56 | | 1.63 | | | |
| | (1.07, 2.28) | | (1.02, 2.60) | | | |
| **Been spat upon** | 1.55 | | | 1.75 | | |
| | (1.04, 2.31) | | | (1.08, 2.85) | | |
| **Victimization (yes vs. no)** | 2.02 | | | | 2.31 | |
| | (1.29, 3.16) | | | | (1.38, 3.87) | |

*Multivariable models adjusted for age, sexual orientation, mother's education, father's education, health literacy, race, and ethnicity.

Weekly Report (MMWR), where 21% and 8% of SGM high school students reported riding with an intoxicated driver and driving a car while under the influence, respectively [9]. We also demonstrated that riding with an intoxicated driver or driving a car while under the influence was associated with homophobic victimization. Taken together, we contend that victimization has negative consequences on the health and behavior of adolescent MSM and is an important public health issue that should be addressed.

Feeling unsafe at school because of one's sexual orientation has been linked to compromised academic achievement, school absenteeism, aggressive behavior, and adverse health outcomes [18, 22, 31]. For this reason, increasing recognition of victimization as a public health issue has led to the implementation of anti-bullying policies that have been shown to effectively reduce violence in schools and improve various outcomes [33, 34]. In fact, some anti-bullying policies encourage teachers to be more supportive of SGM youth, which has shown to buffer the effects of victimization on underage alcohol use [35]. However, these policies are limited to the school environment and do very little to address homophobic victimization occurring in the community or other settings [9]. Because some SGM youth continue to grabble with growing up in communities that may not approve of their sexual orientation [25], creating more supportive community environments coupled with efforts that promote the inclusion of SGM youth in the community is critical. Such efforts may reduce homophobic victimization and yield considerable benefits to alcohol-related prevention programs for SGM youth. Yet, more research is needed to identify both effective and culturally appropriate strategies.

There were several limitations to the study findings. First, homophobic victimization, alcohol use, and alcohol-related behaviors were all self-reported by participants. Consequently, the study findings linking homophobic victimization to underage drinking and alcohol-related behaviors were subject to recall and social desirability bias [36, 37]. Second, there were

**Table 5. Bivariate and multivariable models assessing the association between exposure to homophobic victimization and riding with an intoxicated driver or driving while under the influence of alcohol or drugs.**

| | Riding with an intoxicated driver or driving while under the influence | | | | | |
|---|---|---|---|---|---|---|
| | Bivariate | Multivariable* | Multivariable* | Multivariable* | Multivariable* | Multivariable* |
| | OR (95% CI) | OR (95% CI) | OR (95% CI) | OR (95% CI) | OR (95% CI) | OR (95% CI) |
| **Verbally insulted** | 1.83 | 2.08 | | | | |
| | (1.18, 2.82) | (1.28, 3.39) | | | | |
| **Threatened with physical violence** | 1.37 | | 1.22 | | | |
| | (1.01, 1.86) | | (0.86, 1.73) | | | |
| **Had an object thrown at you** | 1.46 | | | 1.35 | | |
| | (1.06, 2.00) | | | (0.93, 1.97) | | |
| **Been punched, kicked, or beaten** | 1.58 | | | | 1.60 | |
| | (1.12, 2.23) | | | | (1.06, 2.41) | |
| **Attacked sexually** | 1.74 | | | | | 1.83 |
| | (1.21, 2.51) | | | | | (1.20, 2.77) |
| **Someone threatened to out you** | 1.83 | 2.05 | | | | |
| | (1.34, 2.51) | (1.43, 2.93) | | | | |
| **Someone chased or followed you** | 1.64 | | 1.59 | | | |
| | (1.15, 2.34) | | (1.05, 2.41) | | | |
| **Your property was damaged** | 1.62 | | | 1.66 | | |
| | (1.11, 2.37) | | | (1.06, 2.60) | | |
| **Been spat upon** | 1.61 | | | | 1.74 | |
| | (1.08, 2.41) | | | | (1.09, 2.76) | |
| **Victimization (yes vs. no)** | 2.17 | | | | | 2.25 |
| | (1.28, 3.67) | | | | | (1.26, 4.00) |

*Multivariable models adjusted for age, sexual orientation, mother's education, father's education, health literacy, race, and ethnicity.

limitations to some of our measures. For instance, we were unable to determine where homophobic victimization occurred and could not control for school-based victimization in our analysis. Also, when assessing riding or driving while under the influence, we did not differentiate between the use of alcohol or drugs, and the frequency or intensity of alcohol consumption was not measured. Third, the cross-sectional design of the study limited our ability to assess temporality of the associations, and causal inferences could not be made. Finally, we did not estimate the prevalence of homophobic victimization over time, nor did we determine how often these events occurred throughout a person's lifetime. Future studies should examine the cumulative effects of multiple homophobic victimization events over time to better understand the long-term impacts on the health and behavior of adolescent MSM.

This study also had some notable strengths. First, it expanded earlier work examining the association between homophobic victimization and alcohol-related outcomes with a focused analysis on adolescent MSM [3, 26, 38]. Using data from the MyPEEPS study, we had an unprecedented opportunity to investigate alcohol-related outcomes and various homophobic victimization events among a nationwide sample of adolescent MSM. For further analysis, we also included data on risky alcohol-related behaviors, such as riding with an intoxicated driver or driving a car while under the influence of alcohol or drugs. Finally, adolescent MSM may face rejection, isolation, verbal harassment, and physical violence at school, home, or other settings [25], yet most studies have focused on school-based victimization only [31]. Our study adds to the existing body of literature by examining general homophobic victimization as well as various forms of homophobic victimization experienced by adolescent MSM.

The extent to which adolescent MSM engage in underage drinking is concerning due to its immediate and long-term effects on adolescent health and behavior [11, 39]. Due to homophobic victimization experienced by adolescent MSM, they are at increased risk of alcohol misuse [23]. This study highlights the importance of reducing homophobic victimization and developing alcohol-related intervention strategies tailored to this population. By identifying potential risk factors associated with alcohol-related outcomes, we hope that the results of this study will be used to bring awareness to the high prevalence of homophobic victimization and risky alcohol-related behaviors among adolescent MSM.

## Supporting information

**S1 Table. Percent change in odds ratio of multivariable models to assess change in the association between homophobic victimization and underage drinking among adolescent MSM.**
(DOCX)

**S2 Table. Percent change in odds ratio of multivariable models to assess change in the association between homophobic victimization and riding with an intoxicated driver or driving while under the influence among adolescent MSM.**
(DOCX)

**S1 Data.**
(XLS)

## Author Contributions

**Conceptualization:** Robert Garofalo, Lisa M. Kuhns, Cynthia Pearson, D. Scott Batey, Rebecca Schnall.

**Data curation:** Rebecca Schnall.

**Formal analysis:** Rebecca Schnall.

**Funding acquisition:** Rebecca Schnall.

**Investigation:** Robert Garofalo, Lisa M. Kuhns, Cynthia Pearson, D. Scott Batey, Asa Radix, Uri Belkind, Marco A. Hidalgo, Sabina Hirshfield, Rebecca Schnall.

**Methodology:** Rebecca Schnall.

**Project administration:** Josh Bruce, Asa Radix, Uri Belkind, Rafael Garibay Rodriguez, Rebecca Schnall.

**Resources:** Rebecca Schnall.

**Software:** Rebecca Schnall.

**Supervision:** Rebecca Schnall.

**Validation:** Rebecca Schnall.

**Visualization:** Rebecca Schnall.

**Writing – original draft:** Evette Cordoba.

**Writing – review & editing:** Robert Garofalo, Lisa M. Kuhns, Cynthia Pearson, Josh Bruce, D. Scott Batey, Asa Radix, Uri Belkind, Marco A. Hidalgo, Sabina Hirshfield, Rafael Garibay Rodriguez, Rebecca Schnall.

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
