## [Decision Letter · Decision Letter 0]

13 May 2021

PONE-D-20-39463

Risk-taking behaviors in adolescent men who have sex with men (MSM): An association between homophobic victimization and alcohol consumption

PLOS ONE

Dear Dr. Cordoba

Thank you for submitting your manuscript to PLOS ONE. After careful consideration, we feel that it has merit but does not fully meet PLOS ONE’s publication criteria as it currently stands. Therefore, we invite you to submit a revised version of the manuscript that addresses the points raised during the review process.

Please be sure to review and respond to the points raised by the reviewers. 

We look forward to receiving your revised manuscript.

Kind regards,

Jill Blumenthal

Academic Editor

PLOS ONE

Journal Requirements:

2. Please include additional information regarding the survey or questionnaire used in the study and ensure that you have provided sufficient details that others could replicate the analyses. For instance, if you developed a questionnaire as part of this study and it is not under a copyright more restrictive than CC-BY, please include a copy, in both the original language and English, as Supporting Information, or include a citation if it has been published previously.

3. In your discussions and conclusions please take care to avoid overstating your findings. Statements implying causality from correlational research need to be revised. For example, avoid the use of terms such as "risk of" or “effects." Instead consistently use terms such as "associated with" or "associations.

4. In statistical methods, please refer to any post-hoc corrections to correct for multiple comparisons during your statistical analyses. If these were not performed please justify the reasons. Please refer to our statistical reporting guidelines for assistance (https://journals.plos.org/plosone/s/submission-guidelines.#loc-statistical-reporting).

Reviewers' comments:

Reviewer's Responses to Questions

**Comments to the Author**

1. Is the manuscript technically sound, and do the data support the conclusions?

Reviewer #1: Partly

Reviewer #2: Yes

2. Has the statistical analysis been performed appropriately and rigorously? 

Reviewer #1: Yes

Reviewer #2: Yes

3. Have the authors made all data underlying the findings in their manuscript fully available?

Reviewer #1: Yes

Reviewer #2: Yes

4. Is the manuscript presented in an intelligible fashion and written in standard English?

Reviewer #1: Yes

Reviewer #2: Yes

5. Review Comments to the Author

Reviewer #1: The data analysis is routine and includes univariate analyses with non parametric approaches such as Fisher’s exact test when appropriate.. Logistic regression models and multivariable logistic regression were applied appropriately. The sample size appears reasonable. The conclusion appears to follow from the analysis in that adolescent men who experience homophobic victimization are at increased risk of alcohol consumption and risky-alcohol related behaviors.

The strengths and weaknesses of the manuscript are noted by the investigators. However, some concordance and sensitivity measures of these models should have been included in the analyses.

Reviewer #2: The purpose of this study was to examine if there was an association between homophobic victimization and two alcohol related outcomes – alcohol consumption and riding with an intoxicated driver/riding in a car while under the influence of alcohol or drugs. The article is methodologically sound and has the potential to contribute a more nuanced understanding of how different forms of homophobic victimization contribute to substance use related outcomes for young men who have sex with men (YMSM). As the authors point out, there have been numerous studies that find this association--the novel contributions of this study are (1) looking at different aspects of homophobic victimization, (2) focusing on YMSM, and (3) looking at a less studied outcome (riding with an intoxicated driver or driving while intoxicated). My main, critique of the study is that these novel contributions are not highly emphasized, and the front end of the paper is highly general rather than specific to the current study, which is critical given numerous other studies with similar findings.

Abstract

Given the cross-sectional nature of the data, the authors should rephrase this sentence, “Adolescent men who experience homophobic victimization are at increased risk of alcohol consumption and risky-alcohol related behaviors.” so that it sounds less directional.

Introduction

On page 5, the authors conclude by saying: “However, results have been inconsistent, including a null finding [24]. More research is needed to better understand risk factors contributing to underage drinking among SGM youth.” Its unclear which results have been inconsistent – either that homophobic victimization is linked to alcohol related behaviors, or that school-based victimization (not clear if the school-based victimization was LGB specific or not) being the strongest predictor is inconsistent—either way, one null finding does not necessarily make the findings inconsistent given how much research has been done on both topics. The authors should consider rephrasing this sentence to provide a stronger conclusion regarding previous research, perhaps linking it specifically why more research is needed for homophobic victimization as a risk factor for underage drinking.

In the last paragraph before the methods the authors clearly outline the gaps of previous research, but do not provide much specific justification for why these particular gaps matter and are too general.

For the first gap, I do not buy that there is sparse research on homophobic victimization outside of school contexts—the Katz-Wise study that the authors cite, addresses several types of non-school-based victimization for LGB people in research studies conducted up to 2012 (which was 9 years ago), and school-based victimization was not the most frequent forms of victimization measured in those studies.

For the second gap—it is true that few studies compare several forms of homophobic victimization, but the authors should explain why comparing different types of homophobic victimization matters for understanding the effect of homophobic victimization on substance use.

For the third gap, there have been plenty of studies on risk factors within SGM groups, the statement is again very general, which make it boarder on being untrue—there are lots of studies on risk factors within SGM groups (MSM in particular), but there are fewer specific to homophobic victimization. The authors should spend more time addressing why this is important for the group they study, what will understanding this association among YMSM tell us that previous studies cannot or do not?

Lastly, one novel contribution of this study is looking at a less common substance use-related outcome-- riding with an intoxicated driver or driving while intoxicated—and the authors do not provide much rationale for examining this additional outcome.

Methods

The methods are solid. One question had that was unaddressed was how the authors dealt with missing data. There doesn’t appear to be much missing on homophobic victimization, but there is some missing on some of the covariates would mean that the regressions would have different sample sizes.

I also noted here that there is a discrepancy with abstract where it says that 88% of youth reported at least one form of homophobic victimization, and in the text it is reported as 87% in text—"Most participants (87%) were exposed to at least one form of homophobic victimization in their lifetime”

Discussion

In the first full paragraph of the discussion, the authors mention several studies and prevalence rates of different types (homophobic?) victimization from other studies, but then conclude by saying, “Caution should be taken when comparing results across studies due to differences in the study populations and the measure of homophobic victimization.” Why mention the prevalence rates then? What does this add to understanding the current study? This space could be better utilized discussing why these different associations inform our understanding risk factors for substance use-related behaviors.

For example, its perhaps less surprising that someone who was attacked sexually had greater odds of risky substance use behaviors because there is a research base supporting this association, but why having one’s property damaged be related to substance use? Along the same lines, its notable that many of the odds ratios are pretty similar—actually, the largest association was for experiencing any victimization, which is not an uncommon way to use a multi-item measure homophobic victimization. So, do the different forms matter if they all have similar effects on the outcomes? Even if this is not discussed in this specific paragraph, this is not addressed at all throughout the discussion, yet it is one of the important and novel contributions of the study.

6. PLOS authors have the option to publish the peer review history of their article (what does this mean?). If published, this will include your full peer review and any attached files.

Reviewer #1: No

Reviewer #2: No

---

## [Author Response · Author response to Decision Letter 0]

12 Jun 2021

Author’s response to reviewers' comments:

We thank the reviewers for their comments and suggestions on our manuscript, entitled “Risk-taking behaviors in adolescent men who have sex with men (MSM): An association between homophobic victimization and alcohol consumption.” We substantially revised our manuscript in response to these suggestions, and we feel that our revised manuscript is stronger because of these changes. Revisions are tracked in the revised manuscript. Detailed responses to reviewers are as follows (see bold font):

Reviewer #1: 

The data analysis is routine and includes univariate analyses with non parametric approaches such as Fisher’s exact test when appropriate. Logistic regression models and multivariable logistic regression were applied appropriately. The sample size appears reasonable. The conclusion appears to follow from the analysis in that adolescent men who experience homophobic victimization are at increased risk of alcohol consumption and risky-alcohol related behaviors.

1. The strengths and weaknesses of the manuscript are noted by the investigators. However, some concordance and sensitivity measures of these models should have been included in the analyses. 

Thank you for this suggestion. However, we believe that concordance measures are not appropriate to compare the models in our study because such measures are most often used to assess the discriminative ability of risk prediction models, whereas our analysis included causal models. Additionally, a sensitive analysis is used to compare models with varying modelling methods of analysis, different definitions of outcomes, or assess the effects of protocol deviations, missing data, or outliers. This was not the case in our study; all models used the same method of analysis and outcome definition but had different exposure variables (i.e., varies forms of victimization). Instead, to accurately compare these models to one another we compared the effect measures of each model with specific forms of victimization to the effect measure of the generalized victimization model (those experiencing any form of victimization). This analysis has been added as supplementary tables on page 25 of the manuscript.

Reviewer #2: 

1. The purpose of this study was to examine if there was an association between homophobic victimization and two alcohol related outcomes – alcohol consumption and riding with an intoxicated driver/riding in a car while under the influence of alcohol or drugs. The article is methodologically sound and has the potential to contribute a more nuanced understanding of how different forms of homophobic victimization contribute to substance use related outcomes for young men who have sex with men (YMSM). As the authors point out, there have been numerous studies that find this association--the novel contributions of this study are (1) looking at different aspects of homophobic victimization, (2) focusing on YMSM, and (3) looking at a less studied outcome (riding with an intoxicated driver or driving while intoxicated). My main, critique of the study is that these novel contributions are not highly emphasized, and the front end of the paper is highly general rather than specific to the current study, which is critical given numerous other studies with similar findings.

Thank you for this comment. We have revised the language in third paragraph of the introduction to emphasize the novel contributions of this study to the current literature, as suggested. 

2. Abstract: Given the cross-sectional nature of the data, the authors should rephrase this sentence, “Adolescent men who experience homophobic victimization are at increased risk of alcohol consumption and risky-alcohol related behaviors.” so that it sounds less directional.

This sentence has been revised to read: Increased risk of alcohol consumption and risky alcohol-related behaviors were found among adolescent men who experienced homophobic victimization.

3. Introduction: On page 5, the authors conclude by saying: “However, results have been inconsistent, including a null finding [24]. More research is needed to better understand risk factors contributing to underage drinking among SGM youth.” Its unclear which results have been inconsistent – either that homophobic victimization is linked to alcohol related behaviors, or that school-based victimization (not clear if the school-based victimization was LGB specific or not) being the strongest predictor is inconsistent—either way, one null finding does not necessarily make the findings inconsistent given how much research has been done on both topics. The authors should consider rephrasing this sentence to provide a stronger conclusion regarding previous research, perhaps linking it specifically why more research is needed for homophobic victimization as a risk factor for underage drinking.

Thank you for this suggestion. We have revised this sentence to read: Future research is needed to identify the specific forms of homophobic victimization experienced by SGM youth associated with increased risk of alcohol consumption for a comprehensive understanding of risk factors contributing to underage drinking in this population.

4. Introduction: In the last paragraph before the methods the authors clearly outline the gaps of previous research, but do not provide much specific justification for why these particular gaps matter and are too general.

Justification for studying specific homophobic victimization has been mentioned as: Such information is also essential to fully capture the homophobic victimization experiences of SGM youth and how these experiences contribute to the risk of underage drinking for the implementation of future interventions. 

Justification for studying adolescent MSM only has been mentioned as: Research on underage alcohol consumption among adolescent MSM is needed to improve our understanding of alcohol-related disparities found within the SGM population.

Justification for study the association between homophobic victimization and driving while intoxicated or riding with an intoxicated driver has been added to the manuscript as: Previously, over 20% of SGM youth reported riding with an intoxicated driver and 8% drove while intoxicated [9]. Identifying homophobic victimization as a risk factor for alcohol-related behaviors may inform future studies and provide evidence that victimization can influence adverse health behaviors beyond substance abuse, and significantly impact on the health of adolescent MSM.

5. Introduction: For the first gap, I do not buy that there is sparse research on homophobic victimization outside of school contexts—the Katz-Wise study that the authors cite, addresses several types of non-school-based victimization for LGB people in research studies conducted up to 2012 (which was 9 years ago), and school-based victimization was not the most frequent forms of victimization measured in those studies.

The first gap mentioned has been removed from the introduction, as suggested. 

6. Introduction: For the second gap—it is true that few studies compare several forms of homophobic victimization, but the authors should explain why comparing different types of homophobic victimization matters for understanding the effect of homophobic victimization on substance use.

As suggested, we added a sentence to emphasize the importance of studying several forms of homophobic victimization. This sentence reads: By identifying which form of homophobic victimization is associated with alcohol consumption, targeted interventions can be implemented to end specific victimization tactics aimed at SGM youth.

7. Introduction: For the third gap, there have been plenty of studies on risk factors within SGM groups, the statement is again very general, which make it boarder on being untrue—there are lots of studies on risk factors within SGM groups (MSM in particular), but there are fewer specific to homophobic victimization. The authors should spend more time addressing why this is important for the group they study, what will understanding this association among YMSM tell us that previous studies cannot or do not?

As suggested, we added a few sentences to address why it is important to study the association between homophobic victimization and alcohol consumption among YMSM. These sentences read: Second, little research has been conducted to understand victimization as a risk factor for alcohol use specifically for adolescent men who have sex with men (MSM) [3]. We contend that adolescent MSM may be at particularly high risk of underage drinking due to the severe forms of victimization more often reported by them compared to other SGM subgroups [18],[20], warranting concern and future research. Additionally, by focusing on adolescent MSM alone, researchers will be able to compare alcohol consumption among adolescent MSM who experienced homophobic victimization to adolescent MSM who did not, whereas previous comparisons have been made to heterosexual youth [2], [6], [8].

8. Introduction: Lastly, one novel contribution of this study is looking at a less common substance use-related outcome-- riding with an intoxicated driver or driving while intoxicated—and the authors do not provide much rationale for examining this additional outcome. 

We have revised the language in the introduction to provide more rationale for why we examined the outcome ‘riding with an intoxicated driver or driving while intoxicated’. We added the following sentences: Third, to our knowledge, no previous study has assessed the association between homophobic victimization and risky alcohol-related behaviors, specifically driving while intoxicated or riding with an intoxicated driver. Previously, over 20% of SGM youth reported riding with an intoxicated driver and 8% drove while intoxicated [9]. Identifying homophobic victimization as a risk factor for alcohol-related behaviors may inform future studies and provide evidence that victimization can influence adverse health behaviors beyond substance abuse, and significantly impact on the health of adolescent MSM.

9. Methods: The methods are solid. One question had that was unaddressed was how the authors dealt with missing data. There doesn’t appear to be much missing on homophobic victimization, but there is some missing on some of the covariates would mean that the regressions would have different sample sizes.

Thank you for pointing out this omission. We added the sentence “Participants with missing data on covariates were dropped from the multivariable regression models” to clarify how missing data was handled in the regression models on page 9, line 199-200. 

10. Methods: I also noted here that there is a discrepancy with abstract where it says that 88% of youth reported at least one form of homophobic victimization, and in the text it is reported as 87% in text—"Most participants (87%) were exposed to at least one form of homophobic victimization in their lifetime”

This sentence has been corrected in the abstract and now reads: Most participants (87%) reported at least one form of homophobic victimization in their lifetime, with verbal insults being the most frequently reported (82%). This reflects the data presented in text and in Table 2. 

11. Discussion: In the first full paragraph of the discussion, the authors mention several studies and prevalence rates of different types (homophobic?) victimization from other studies, but then conclude by saying, “Caution should be taken when comparing results across studies due to differences in the study populations and the measure of homophobic victimization.” Why mention the prevalence rates then? What does this add to understanding the current study? This space could be better utilized discussing why these different associations inform our understanding risk factors for substance use-related behaviors. For example, its perhaps less surprising that someone who was attacked sexually had greater odds of risky substance use behaviors because there is a research base supporting this association, but why having one’s property damaged be related to substance use? Along the same lines, its notable that many of the odds ratios are pretty similar—actually, the largest association was for experiencing any victimization, which is not an uncommon way to use a multi-item measure homophobic victimization. So, do the different forms matter if they all have similar effects on the outcomes? Even if this is not discussed in this specific paragraph, this is not addressed at all throughout the discussion, yet it is one of the important and novel contributions of the study.

Thank you for this comment. The first full paragraph on prevalence rates has been removed from the discussion, as suggested. A discussion on the different associations and how they inform our understanding of risk factors for substance use-related behaviors has been added on page 20, line 319‒331. We also addressed the finding that the different forms of victimization have similar effect on the outcomes in order to increase clarity of its significance.

---

## [Decision Letter · Decision Letter 1]

22 Jul 2021

PONE-D-20-39463R1

Risk-taking behaviors in adolescent men who have sex with men (MSM): An association between homophobic victimization and alcohol consumption

PLOS ONE

Dear Dr. Cordoba,

Thank you for submitting your manuscript to PLOS ONE. After careful consideration, we feel that it has merit but does not fully meet PLOS ONE’s publication criteria as it currently stands. Therefore, we invite you to submit a revised version of the manuscript that addresses the points raised during the review process.

We look forward to receiving your revised manuscript.

Kind regards,

Jill Blumenthal

Academic Editor

PLOS ONE

Journal Requirements:

Reviewers' comments:

Reviewer's Responses to Questions

**Comments to the Author**

1. If the authors have adequately addressed your comments raised in a previous round of review and you feel that this manuscript is now acceptable for publication, you may indicate that here to bypass the “Comments to the Author” section, enter your conflict of interest statement in the “Confidential to Editor” section, and submit your "Accept" recommendation.

Reviewer #1: All comments have been addressed

Reviewer #2: (No Response)

2. Is the manuscript technically sound, and do the data support the conclusions?

Reviewer #1: (No Response)

Reviewer #2: Yes

3. Has the statistical analysis been performed appropriately and rigorously? 

Reviewer #1: (No Response)

Reviewer #2: Yes

4. Have the authors made all data underlying the findings in their manuscript fully available?

Reviewer #1: (No Response)

Reviewer #2: Yes

5. Is the manuscript presented in an intelligible fashion and written in standard English?

Reviewer #1: (No Response)

Reviewer #2: No

6. Review Comments to the Author

Reviewer #1: (No Response)

Reviewer #2: The authors have generally responded well to my comments. I have a few remaining suggestions listed below.

• I recommend that the authors update their literature review and focus it only on research specific to MSM or SGM men. Much the research on SGM populations related to alcohol use and substance use is discussed in terms of gender difference among SGM populations, so general findings about SGM don’t necessarily apply to this study focused on MSM. I have more specific suggestions below, but the discussion should also be focused more around MSM than SGM populations more generally.

• I noted some typographical errors in the literature review, so the authors should carefully edit the manuscript.

• The authors have a claim that school-based homophobic victimization is one of the strongest predictors of alcohol use on page 5 line 95 and 96. However, neither of the references cited makes this claim and I am unsure that that there has been a study testing this claim. I recommend the authors remove the statement or update to say that school-based victimization has been fond to be strongly related to substance use.

• The discussion still has track changes in it.

• In the discussion, the authors state: “We found that half of the adolescent MSM study participants reported alcohol consumption, which is noteworthy since they are all under the US legal drinking age of 21. However, this estimate was lower than that previously reported by a nationwide study, where 75% of SGM high school students drank alcohol [9]. This difference may be explained by our younger sample population, which included both middle school and high school students.” This statement should be removed—the comparison is MSM to SGM which while there is overlap are not the same populations, so there are numerous reasons that the prevalence rates could differ beyond those the author’s mention, namely that SGM women tend to have more elevated rates of alcohol use than SGM men.

• In the discussion, the authors state: “We found that among adolescent MSM underage drinking was associated with various forms of homophobic victimizations, such as verbal insults or threats, physical or sexual assaults, and property damage. This finding aligned with results from existing studies focused on school-victimization and alcohol use among SGM youth.” It is unclear why they compare their results to studies on school victimization given that their study focuses on general victimization of MSM.

7. PLOS authors have the option to publish the peer review history of their article (what does this mean?). If published, this will include your full peer review and any attached files.

Reviewer #1: No

Reviewer #2: No

---

## [Author Response · Author response to Decision Letter 1]

24 Aug 2021

Editor and Reviewer comments:    

We thank the reviewers for their comments and suggestions on our manuscript, entitled “Risk-taking behaviors in adolescent men who have sex with men (MSM): An association between homophobic victimization and alcohol consumption.” We substantially revised our manuscript in response to these suggestions, and we feel that our revised manuscript is stronger because of these changes. Revisions are tracked in the revised manuscript. Detailed responses to reviewers are as follows (see bold text below):

Review Comments to the Author

Reviewer #1: (No Response)

Reviewer #2: The authors have generally responded well to my comments. I have a few remaining suggestions listed below.

Reviewer #2 Comments:

1. I recommend that the authors update their literature review and focus it only on research specific to MSM or SGM men. Much of the research on SGM populations related to alcohol use and substance use is discussed in terms of gender difference among SGM populations, so general findings about SGM don’t necessarily apply to this study focused on MSM. I have more specific suggestions below, but the discussion should also be focused more around MSM than SGM populations more generally.

Thank you for this thoughtful comment. While we agree with the reviewer that much of the research on SGM populations related to alcohol use and substance use is discussed in terms of gender difference among SGM populations, our literature review was limited by the current lack of research on alcohol use among adolescent MSM. Because this study focused on underage drinking among adolescent MSM, we believe that a comparison to older MSM (i.e., 21 years or older) may not highlight the risk of this behavior. Instead, we aimed to compare our results to other youth who may be at risk of underage drinking (i.e., SGM youth). 

2. I noted some typographical errors in the literature review, so the authors should carefully edit the manuscript.

We thoroughly reviewed the full manuscript for typographical errors. The appropriate edits have been made to the manuscript. 

3. The authors have a claim that school-based homophobic victimization is one of the strongest predictors of alcohol use on page 5 line 95 and 96. However, neither of the references cited makes this claim and I am unsure that that there has been a study testing this claim. I recommend the authors remove the statement or update to say that school-based victimization has been found to be strongly related to substance use.

As suggested by the reviewer, we revised the sentence to read “In fact, some studies have found school-based victimization to be strongly related to substance use among SGM youth [3], [15].”

4. The discussion still has track changes in it.

We apologize for this oversight. For this submission, we submitted two versions of the manuscript: 1. Manuscript with tracked changes and 2. Manuscript with no tracked changes (clean version). 

5. In the discussion, the authors state: “We found that half of the adolescent MSM study participants reported alcohol consumption, which is noteworthy since they are all under the US legal drinking age of 21. However, this estimate was lower than that previously reported by a nationwide study, where 75% of SGM high school students drank alcohol [9]. This difference may be explained by our younger sample population, which included both middle school and high school students.” This statement should be removed—the comparison is MSM to SGM which while there is overlap are not the same populations, so there are numerous reasons that the prevalence rates could differ beyond those the author’s mention, namely that SGM women tend to have more elevated rates of alcohol use than SGM men.

Thank you for your suggestion, this statement has been removed. 

6. In the discussion, the authors state: “We found that among adolescent MSM underage drinking was associated with various forms of homophobic victimizations, such as verbal insults or threats, physical or sexual assaults, and property damage. This finding aligned with results from existing studies focused on school-victimization and alcohol use among SGM youth.” It is unclear why they compare their results to studies on school victimization given that their study focuses on general victimization of MSM.

This is a good point made by the reviewer. While we agree with the reviewer that our study focuses on general victimization among adolescent MSMS, the results of our study were compared to studies on school-based victimization because this is the data currently available. We revised this sentence to read, “We compared our results to studies focused on school-based victimization and alcohol use among SGM youth due to the lack of data surrounding homophobic victimization in MSM youth in other setting.”

---

## [Editor Report · Decision Letter 2]

3 Nov 2021

Risk-taking behaviors in adolescent men who have sex with men (MSM): An association between homophobic victimization and alcohol consumption

PONE-D-20-39463R2

Dear Dr. Evette Cordoba-

We’re pleased to inform you that your manuscript has been judged scientifically suitable for publication and will be formally accepted for publication once it meets all outstanding technical requirements.

Kind regards,

Jill Blumenthal

Academic Editor

PLOS ONE

Additional Editor Comments (optional):

All revisions have been addressed and the manuscript has been accepted for submission.
---

## [Editor Report · Acceptance letter]

17 Nov 2021

PONE-D-20-39463R2 

Risk-taking behaviors in adolescent men who have sex with men (MSM): An association between homophobic victimization and alcohol consumption 

Dear Dr. Cordoba:

I'm pleased to inform you that your manuscript has been deemed suitable for publication in PLOS ONE. Congratulations! Your manuscript is now with our production department. 

Kind regards, 

on behalf of

Dr. Jill Blumenthal 

Academic Editor

PLOS ONE